# A Genome-Wide View of Transcriptional Responses during *Aphis glycines* Infestation in Soybean

**DOI:** 10.3390/ijms21155191

**Published:** 2020-07-22

**Authors:** Luming Yao, Biyun Yang, Xiaohong Ma, Shuangshuang Wang, Zhe Guan, Biao Wang, Yina Jiang

**Affiliations:** 1School of Agriculture and Biology, Shanghai Jiao Tong University, Shanghai 200240, China; lmyao@sjtu.edu.cn (L.Y.); maxiaohong@sjtu.edu.cn (X.M.); 2School of Life Sciences, East China Normal University, Shanghai 200241, China; yangbiyun1996@163.com (B.Y.); flora_wang1004@163.com (S.W.); nov16th@163.com (Z.G.)

**Keywords:** *Glycine max*, *Aphis glycine*, plant–insect interaction, antibiosis, antixenosis

## Abstract

Soybean aphid (*Aphis glycines* Matsumura) is one of the major limiting factors in soybean production. The mechanism of aphid resistance in soybean remains enigmatic as little information is available about the different mechanisms of antibiosis and antixenosis. Here, we used genome-wide gene expression profiling of aphid susceptible, antibiotic, and antixenotic genotypes to investigate the underlying aphid–plant interaction mechanisms. The high expression correlation between infested and non-infested genotypes indicated that the response to aphid was controlled by a small subset of genes. Plant response to aphid infestation was faster in antibiotic genotype and the interaction in antixenotic genotype was moderation. The expression patterns of transcription factor genes in susceptible and antixenotic genotypes clustered together and were distant from those of antibiotic genotypes. Among them APETALA 2/ethylene response factors (AP2/ERF), v-myb avian myeloblastosis viral oncogene homolog (MYB), and the transcription factor contained conserved WRKYGQK domain (WRKY) were proposed to play dominant roles. The jasmonic acid-responsive pathway was dominant in aphid–soybean interaction, and salicylic acid pathway played an important role in antibiotic genotype. Callose deposition was more rapid and efficient in antibiotic genotype, while reactive oxygen species were not involved in the response to aphid attack in resistant genotypes. Our study helps to uncover important genes associated with aphid-attack response in soybean genotypes expressing antibiosis and antixenosis.

## 1. Introduction

Soybean aphid, *Aphis glycine* (Matsumura), is an asexually-reproducing soybean pest native to eastern Asia, and, after its accidental introduction in 2000, soybean aphid became invasive throughout Midwestern U.S. and southern Canada because of the rapid population expansion, high mobility and few limits to migration [1]. Soybean aphid infestation reduces soybean yield by removing photosynthates and inhibiting photosynthesis up to 50% on infested leaves, even without apparent symptoms of aphid injury [2]. Besides, virus transmission by aphids causes plant stunting and leaf deformation, whereas honeydew secreted after injury by aphids facilitates sooty mold growth and infection by soybean cyst nematode [1,2].

Different strategies have been developed to manage soybean aphid infestation. Application of broad-spectrum insecticides is the primary approach to control this pest [3]. However, insecticides may also impact beneficial insects and their repeated application is likely to cause pesticide resistance [4]. An alternative strategy for aphid control is to utilize the existing host-plant resistance in soybean germplasm. Plant defenses against aphids can be categorized as antibiosis, antixenosis, and tolerance. Antibiotic mechanisms influence aphid physiology, such as development time, survival, and fecundity by toxic secondary metabolites, protease inhibitors, and antibiotic effects; antixenotic factors affect aphid behavior, such as plant choice and feeding behavior by volatiles and physical barrier of plant; and plant tolerance allows the plants to harbor a large number of aphids without a significant loss in yield [5,6]. Soybean germplasms exhibiting antibiosis, antixenosis, and tolerance have already been identified [7,8,9,10,11,12,13]. Several resistance genes were mapped in some of those germplasms, such as *Rag1*, *Rag2*, and *Rag_P746*, which confers antibiosis [14,15,16], and *Rag3*, *Rag4*, and *Rag_P203*, encoding for antixenotic resistance [7,8,16]. The mechanics contributing to the resistance in legumes include structural defenses, secondary metabolites, and signaling components [6], but the underlying genetic basis of the aphid–plant interaction is still not well understood [17,18,19].

High throughput technology, including microarray and transcriptome analysis, has been successfully utilized to investigate global responses of plants. Microarray analyses of plants harbored *Rag1* referred to as antibiosis identified 140 genes with specific responses related to resistance when compared with a susceptible soybean cultivar 6 and 12 h post-application of aphids and confirmed an earlier and increased induction of three defense-related genes in aphid-resistant soybean cultivars [20]. Microarray analysis of the long-term responses to aphid of *Rag1*-related resistance in soybean showed that many defense-related transcripts are constitutively expressed in resistant plants [21]; specifically, isoflavonoid biosynthesis is highly related to resistance response [19]. RNA-sequencing (RNA-seq) technology has been applied to query plant responses to aphid infestation. Transcriptome analysis of soybean tolerance to aphids showed that 3 and 36 genes, including genes related to plant defense, were differentially expressed 5 d and 15 d after aphid infestation, respectively [18]. Besides, a candidate *R* gene controlling aphid resistance was discovered with RNA-seq transcriptome analysis in antibiotic (*Rag2*) soybean [22]. However, the differences of global responses between antibiotic and antixenotic genotypes in soybean are still uncovered.

In the present study, we performed transcriptome profiling of the responses in aphid resistance soybean genotypes P203 (antixenosis) and P746 (antibiosis) 24 h, 48 h, and 96 h after aphid infestation and conducted a comprehensive analysis of the genotypes to identify the dynamic alteration of gene expression. This study will contribute to the understanding of aphid-resistant mechanisms underlying different plant defense catalogs.

## 2. Results

### 2.1. Global Analysis of Gene Expression after Aphid Infestation

To obtain the global gene expression profiles during aphid infestation in soybean genotypes, samples with three biological replicates at 24, 48, and 96 h after incubation (HAI) in the susceptible Dongnong47, antixenotic P203, and antibiotic P746 genotypes were collected. A total of 18 libraries were constructed and approximately 990 million raw reads were obtained, with approximately 53.1 million clean reads per library. About 911 million mapped reads were generated, with approximately 48.5 million uniquely mapped reads per library (Appendix A). The number of clean data comprised approximately 96.65% of the total number of raw data. The distribution of clean reads on chromosomes was analyzed to estimate the distribution of genes responding to aphid infestation (Appendix A) according to the latest soybean genome Wm82.a2.v1 from the Ensembl Plants database (https://plants.ensembl.org/index.html). Most of the clean reads mapped to chromosomes 13, 19, and 8 in all genotypes (Dongnong47: 89,451,410 reads, 25.77%; P203: 83,081,860 reads, 25.62%; P746: 76,451,398 reads, 23.83%) (Appendix A). In the database, genome assembly contains 55,897 annotated genes, and a total of 672,566 annotated genes were obtained in all 18 libraries, with the average of 37,365 annotated genes per library (Figure 1A).

The expression levels of annotated genes, which were normalized as transcripts per million (TPM) reads, were generally comparable in all genotypes with aphid infestation (Figure 1B, Appendix A). In each genotype, the expression levels of any two libraries were highly correlated (the average correlation coefficient *r* > 0.93 (Dongnong47: 0.95; P203: 0.94; P746: 0.93) in each genotype (Figure 1C). High correlation was observed even between the expression profiles of aphid infestation and mock treatment, despite the distinct phenotype of the seedlings with aphid infestation. It suggested that only a small set of genes were involved in the response of plant during aphid infestation, while the expression profiles of most genes remain stable.

### 2.2. Dynamic Expression of Differentially Expressed Genes in Response to Aphid Infestation

The expression patterns of genes with aphid infestation in different genotypes were evaluated by DESeq2 (*p*-adjust < 0.05 and |log_2_FC| ≥ 1). In total, 12,573 genes were identified as differentially expressed genes (DEGs) in all libraries, with 5788 DEGs in Dongnong47, 1388 DEGs in P203 and 8221 DEGs in P746 (Appendix A), and, in all genotypes, more than 60% genes not detected at 24 HAI were significantly regulated at 48 HAI and 96 HAI, which indicated lasting response in soybean to biotic stress and more systematic response in susceptible and antibiotic genotypes during aphid infestation (Figure 2A). Besides, at 24 HAI more than 90% DEGs in Dongnong47 and P203 were overlapped with significantly up-regulated, while the number of up- and down-regulated genes were comparable in other DEGs in all genotypes (Figure 2B,C).

In order to investigate possible function of DEGs in different genotypes, the gene ontology (GO) and Kyoto Encyclopedia of Genes and Genomes (KEGG) enrichment analysis was conducted. Overall, enriched GO terms were distinct in different genotypes, indicating divergent responses to aphid infestation (Appendix A). Within the top 10 significantly enriched GO terms (FDR < 0.05), terms related to biological process regulation was overrepresented at 24 HAI and terms associated to phosphorylation and biosynthetic process were enriched at 48 and 96 HAI in Dongnong47; in P203, terms about to homeostasis were significantly enriched at 24 HAI, and genes associated to catabolic process and transport were overrepressive at 96 HAI, but there were no GO terms significantly enriched at 48 HAI; in P746, terms related to regulation were enriched at 24 HAI, followed by biosynthetic and metabolic process at 48 and 96 HAI (Appendix A). The enriched GO terms suggested that transcription regulators were involved and play key roles during aphid infestation. Besides, genes associated to phytohormones regulation (such as jasmonic acid (JA) and salicylic acid (SA) biosynthetic and metabolic process), defense response related to callose (such as callose localization and deposition in cell wall), and superoxide metabolism (such as superoxide radicals removal) were also enriched (Appendix A). With KEGG enrichment analysis, pathways related to plant hormone signaling were significantly enriched in Dongnong47 at 24 and 48 HAI, while there was no significant enriched pathway in P203 at the same time, and in P746 the pathways associated to hormone signaling and secondary metabolite were enriched during aphid infestation (Appendix A). The GO and KEGG enrichment analysis suggested that the similar biological process involved in response to aphid infestation in susceptible and antibiotic genotypes in 24 h, but it was diverse between susceptible and resistant genotypes at 48 and 96 HAI.

### 2.3. Differentially Expressed Transcription Factors in Response to Aphid Infestation

During the plant’s response to biotic stress, transcription factors (TFs) play critical roles in controlling signal initiation, signal transduction, and plant defense by regulating gene expression [23]. The enrichment of GO terms related to transcription regulator observed in the present study indicates that TFs also play important roles in plant’s response to aphid infestation. A total of 894 TF genes categorized into 40 families were identified with distinct expression patterns (Appendix A; Figure 3A), and 43.07% TFs belong to the families of APETALA 2/ethylene response factors (AP2/ERF), bHLH, the transcription factors contained conserved WRKYGQK domain (WRKY), and v-myb avian myeloblastosis viral oncogene homolog (MYB), which indicated they played major roles in regulation of plant response to aphid infestation (Appendix A). The regulation tendency between susceptible and resistant genotypes were opposite. The stress response in Dongnong47 were drastically regulated at 24 HAI and suppressed later, while, in P203 and P746, the number of TFs involved in the response continuing increased. It suggested continuous regulation in resistant genotypes (Appendix A).

The expression patterns of TFs were investigated in all libraries, and k-mean clustering based on the expression patterns of the TF gene families resolved six clusters. Genes in Cluster 1 were highly expressed in aphid-infected Dongnong47 at 48 and 96 HAI with an enrichment of APETALA 2/ethylene response factors (AP2/ERF), basic helix-loop-helix (bHLH), MYB, and WRKY TF genes; genes in Cluster 2, 3, and 4 were highly expressed in P746 and gradually increased during the infestation; and genes in Cluster 5 and 6 were highly expressed in P203 with an enrichment of AP2/ERF TF genes (Figure 3B).

Each cluster obtained by k-mean clustering included different TF families, and each family was present with at least one member in most clusters (Figure 3B). Compared with other families, MYB and WRKY families were enriched in all clusters and exhibited similar changes in gene expression in the three genotypes during infestation, whereas the expression of the AP2/ERF gene family was the same in Dongnong47 and P746, but followed the opposite trend in P203 (Figure 3B).

The TF genes in the AP2/ERF family were clustered into two groups according to their expression patterns with hierarchical clustering. In Group 1, the expression of the genes increased in P203 during infestation, and in Dongnong47 and P746, the genes were up-regulated at 48 HAI and down-regulated at 96 HAI; the down-regulation was greater in Dongnong47. The TFs in Group 2 shared the same decreasing expression tendency during infestation in all genotypes (Figure 4A). The WRKY family of TFs was divided into three groups based on their expression patterns. Dongnong47 and P746 shared the same expression pattern in Group 1 and Group 2, whereas the genes in Group 3 had a different expression tendency in all genotypes (Figure 4B). Clustering of the MYB family of TFs revealed three groups based on the expression patterns. Group 1 genes were up-regulated in all genotypes at 24 HAI and 48 HAI, but their expression gradually decreased in Dongnong47 and P746 at 96 HAI. Group 2 genes were predominantly expressed in P746. In contrast, the genes in Group 3 were dramatically repressed by aphid infestation in P746 (Figure 4C). The expression patterns were confirmed by qPCR in *Glyma14G34590 (GmRAP2)*, *Glyma09G06980 (GmWRKY11)*, and *Glyma03G00890 (GmMYB30)*, which were involved in stress response in plant [24,25,26] (Figure 4D).

### 2.4. The Expression Pattern of Jasmonic Acid- and Salicylic Acid-Related Genes during Aphid Infestation

Phytohormones play essential roles in plant response to biotic stress, especially jasmonic acid (JA) and salicylic acid (SA). Both JA and SA biosynthetic and metabolic process were enriched as GO terms and plant hormone signaling pathways enriched in KEGG pathways (Appendix A). Among the detected DEGs, 49 genes were JA-related, including those encoding of protein for biosynthesis, signaling, receptor and response, and 24 genes were SA-related genes, involve in biosynthesis, signaling and regulator (Appendix A). Hierarchical clustering of the genes showed that most of JA-related genes in Dongnong47 and P746 were down-regulated at 96 HAI, while in P203 the genes were up-regulated at the same time point (Figure 5A). And the same expression patterns were also observed in SA-related genes (Figure 5B). The results of the qPCR for *Glyma09G09100* (*GmJAZ12*), a key regulator in the JA signal transduction pathway and *Glyma10G06600* (*GmPAL1*), the key role in SA biosynthesis were consistent with the RNA-seq results (Figure 5C).

To investigate the possible function of the JA- and SA-related genes, the concentrations of JA and SA were analyzed using liquid chromatography-mass spectrometry (LC-MS). The JA concentration was significantly increased in Dongnong47 during infestation, but significantly decreased at 96 HAI in P203 and P746 (*p <* 0.05), which indicated the presence of different regulation mechanisms in susceptible and resistant genotypes (Figure 5D). The SA concentration remained constant after infestation in Dongnong47 and P203 but increased significantly in P746, which suggested that the SA pathway is not involved in the response to aphid invasion in susceptible and antixenotic genotypes but plays an important role in antibiotic resistance of P746 (Figure 5E).

### 2.5. Change in the Expression of Callose Synthase Genes during Aphid Infestation

A total of 15 genes related to callose synthase (CalS) were predicted in SoyBase database (Wm82.a1.v1.1; https://www.soybase.org/), and 9 genes showed detectable expression during aphid infestation in all genotypes (Appendix A). The expression of these genes peaked at 48 HAI in Dongnong47, most of the genes up-regulated in P203 at 96 HAI, and the down-regulation tendency was observed in P746 during aphid infestation (fold change (FC) > 1) (Figure 6A).

To investigate the possible function of callose during aphid infestation, the callose deposition was detected with aniline blue staining. Callose deposition was significantly increased in all genotypes with aphid colonization. And at 96 HAI the highest deposition was observed in Dongnong47, followed by that in P203 and P746. In P746, there was no significant difference in callose deposition between 48 HAI and 96 HAI, but it was greater compared with that in Dongnong47 and P203 at 48 HAI (Figure 6B,C). These results suggested differential callose regulation between antibiotic genotypes and that in susceptible and antixenotic genotypes.

### 2.6. Soybean Response to Reactive Oxygen Species during Aphid Infestation

Reactive oxygen species (ROS) are produced in a plant during its exposure to abiotic and biotic stresses, and their levels are modulated by a catalase. Six catalase-related genes were predicted in SoyBase (Wm82.a1.v1.1), and four genes showed an appreciable expression in RNA-seq dataset (Appendix A). Genes, except *Glyma14g39810*, were repressed in all genotypes during aphid attack and significantly down-regulated (log_2_FC < −1) in Dongnong47 at 96 HAI (Figure 7A). Staining with 3,3′-diaminobenzidine (DAB) applied to determine ROS in plants with aphid colonization revealed no significant difference between plants with or without infestation in P203 and P746 (Figure 7B,C), which indicated that peroxisome metabolism was not involved in plant resistance response to aphids.

## 3. Discussion

Soybean aphids are an invasive species with worldwide distribution. A systematic study of the gene expression regulation in response to soybean aphid infestation helps to uncover the molecular mechanisms of aphid resistance in soybean. Previous studies on the response of soybean to soybean aphid, which used high throughput technology, focused on soybean genotypes with antibiosis harboring *Rag1* and *Rag2* [19,20,21,22]. Here, we revealed transcriptome profiles of both antibiotic and antixenotic genotypes harboring resistant genes [16,27] 24 h, 48 h, and 96 h after aphid infestation. Several results were also observed in resistant or susceptible genotypes of other plants under insect attack or bacterial infection, which indicated the general response of plant to biotic stress.

Different resistant types of soybean differ in their levels of resistance to aphid infestation and survival. Nevertheless, the correlation coefficient was extremely high in each genotype between aphid-infested and mock-treated plants (Figure 1C), and a number of DEGs were found among different genotypes and sampling at different times after infestation (Appendix A). High correlation was reported in the tolerant soybean genotype KS4202 with two DEGs identified under aphid attack 2 days after infestation [18], while the growth of aphid population in *Rag1*-harboring soybean dramatically decelerated without a strong measurable transcriptional response within 7 days after aphid infestation [21]. In *Medicago sativa*, 3919 of 184,892 unigenes generated were differentially expressed in both resistant and susceptible genotypes exposed to aphid infestation [28]. These findings suggested that only a small set of genes are involved in the response to aphid infestation, while the expression profiles of most genes remain stable.

Plant defense to aphid attack was distinct between susceptible and resistant soybeans. Our gene expression profiles demonstrated that the response to aphid infestation in antibiotic resistant genotypes was faster than in the susceptible genotype during early infestation (24 h). Similar results were reported in Dowling (*Rag1*) when compared with Williams 82 (susceptible) after aphid treatment: 104 DEGs were detected in Dowling and 51 DEGs in Williams 82 at 6 HAI, but the number of DEGs in the two genotypes became comparable at 24 HAI [20]. Likewise, resistant tobacco cultivars had more DEGs than did the susceptible cultivar one day after the infection with cucumber mosaic virus [29]. The faster response in antibiotic resistant plants indicated that the plant defense to aphid and other biotic stresses is initiated at early stage of aphid attack and lasts during infestation. However, in antixenotic genotype much smaller set of DEGs involved in the response during aphid invasion, which indicated less interaction between seedlings and aphids.

Transcription factors are key regulators of plant response to aphid infestation. They may be involved in response initiation to aphid infestation and resistance duration (such as AP2/ERF, WRKY, and MYB family of TFs). In our study, the expression patterns of TFs in susceptible and antixenotic genotypes clustered together and were distant from those of antibiotic genotypes (Figure 3B). Antixenotic mechanisms influence aphid behavior, such as plant choice and feeding behavior. Antixenotic plant harbors secondary metabolites that are detrimental to insects and releases volatiles that can repel insects to promote ‘dispersal behavior’ in aphids [30,31]. This means that less impedimental elements are needed in plants and there is less interaction between the plant and the aphid. Antibiotic factors influence aphid physiology, and toxic secondary metabolites affect pest survival [32,33]. Therefore, antixenotic genotypes are more similar to susceptible types in having less special secondary metabolites, whereas antibiotic genotypes are more diversified in resistance interaction between plants and aphids.

In this study, we demonstrated that TF genes in AP2/ERF, WRKY, and MYB families were dramatically regulated in all genotypes under aphid attack (Figure 4). AP2/ERF is a TF family that is considered to function in stress response pathways under both abiotic and biotic stresses [34]. ERFs can either directly regulate the pathogenesis-related gene expression or bind with dehydration-responsive element motif involved in abiotic stress regulation [35]. ERF genes mainly mediate pathogen- and disease-related stimuli by integrating multiple signaling pathways, such as JA, ethylene ET, and SA pathways [36]. Besides, TFs in WRKY and MYB families are also involved in plant defense response to biotic stress, such as response to *Phytophthora parasitica* in tomato [37], resistance response to aphid in soybean and *Chrysanthemum morifolium* [18,38], and response to insect attack and bacterial infection in *Arabidopsis thaliana* [23].

The phytohormones JA and SA contribute to signaling associated with plant–aphid interaction. In our study, a total of 54 JA-related genes and 26 SA-related genes were identified (Figure 5A,B). These results suggested that the JA-responsive pathway was dominant in aphid–soybean interaction. We monitored the JA and SA concentration during aphid infestation and observed that phytohormone accumulation lagged behind the regulation of related genes, which suggested postponement between response initiation and managing the biotic stress during aphid infestation. JA mediated the plant response in susceptible and antixenotic genotypes and SA in antibiotic genotypes was involved in response by suppressing the JA pathway (Figure 5D,E). It is considered that leaf and cell damage caused by leaf-chewing herbivores induces JA, and aphid feeding mostly induces SA [39,40]. Our results pertaining to susceptible and antixenotic genotypes did not corroborate this opinion, but were in agreement with findings reported for susceptible wheat genotypes attacked by greenbug (*Schizaphis graminum*) [41]. This discrepancy may be attributed to different genetic backgrounds of the plants and the interaction mechanisms. The changes in JA and SA concentrations followed an opposite trend in all tested genotypes, suggesting that JA and SA are natural antagonists by inducing one pathway, the other one will be suppressed with hormonal crosstalk [40].

Deposition of callose leads to plugging of sieve plates, which contributes to plant defense against aphids [6]. A massive localized deposition of callose at the feeding site was observed in wheat (*Triticum aestivum*) infested by Russian wheat aphid (*Diuraphis noxia*) [42], which is concurrent with our observations. Callose deposition is stimulated by aphid-derived factors [43,44]. During aphid infestation, particularly in susceptible plants, the presence of saliva and effectors induced greater accumulation of callose at 96 HAI. Besides, callose deposition occurred more rapidly and efficiently after infestation in resistant plants compared with genotype lacking *R* genes [45,46]. Our results demonstrated that significantly more callose deposition was induced in resistant soybean P746 at 48 HAI (Figure 6). The expression pattern of *CalS*, a key gene for callose synthase, was opposite to callose deposition, which suggested that callose deposition is regulated not by a single *CalS* but by a complex of callose synthase genes [47] or other factors [48].

One of the reactions of plants to insect invasion is generation of ROS [49], and their accumulation contributes to plant’s defense against aphids [50,51,52]. However, in our study, although genes related to superoxide metabolism was enriched in GO terms, no significant changes in ROS synthesis were observed in both antixenotic and antibiotic soybean during aphid infestation, but they were increased significantly in the susceptible genotype (Figure 7). Similar results were reported in barley infected by barley yellow dwarf virus (BYDV) [53] and tolerant soybean during a short-term aphid infestation [50]. This suggested that ROS was not involved in the response to aphid attack in resistant soybean with limited infestation period.

## 4. Materials and Methods

### 4.1. Plant Materials and Growth Conditions

Three soybean genotypes with contrasting response to *Aphis glycine* were used in this study: an aphid-susceptible soybean cultivar Dongnong47, the antibiotic genotype P746, and the antixenotic genotype P203 [16,27]. Seeds from each genotype were planted into pots (8 cm deep and 10 cm in diameter) filled with potting media (34% peat, 31% perlite, 31% vermiculite, and 4% soil mix). At least 18 seedlings of each genotype were retained for treatment with one seedling per pot. Seedlings were grown in a chamber at 26 °C with a 16-h photoperiod and 70% relative humidity.

### 4.2. Aphid Culture and Infestation

The aphids were collected from an experimental soybean field of Shanghai Jiao Tong University (E 121°26′, W 31°2′), and the colony was maintained on the susceptible cultivar Dongnong47 under controlled condition with 70% relative humidity, 16-h/8-h light-dark cycle at 26 °C as described previously [54]. In order to synchronizing the age of aphids, apterous viviparae were placed on restricted leaves of susceptible soybean Dongnong47 for 24 h for nymphs, then the third instar nymphs were collected for infestation with desirable aphid survivability

When the first trifoliolate of soybean was fully expanded (V1 stage), each leaflet of the trifoliolate was infested with six aphids using a moist brush. To confine aphid movement, aphids were caged on leaves, and the same cages were applied to the control, mock-treated seedlings, that were not infested with aphids.

The caged areas of leaf samples from both aphid-infested and mock-treated seedlings were harvested at 24 HAI, 48 HAI, and 96 HAI for subsequent analysis.

### 4.3. RNA Extraction and Quantitative Real-Time PCR Validation

Three independent samples for each treatment were harvested for RNA extraction. Total RNA was isolated using an RNAprep Pure Kit (For Plant) with DNase digestion (Tiangen, Beijing, China) (http://www.tiangen.com/en/) according to the manufacturer’s instructions. RNA degradation and contamination were detected by electrophoresis on 1% agarose gels, and its purity and concentration were evaluated with a NanoDrop Lite (Thermo Scientific, Waltham, MA, USA) (https://www.thermofisher.com).

For qPCR validation, three biological replicates, independent of RNA samples, were collected and assayed. FastKing gDNA Dispelling RT SuperMix (Tiangen) was used for reverse transcription. Quantitative real-time PCR was performed in triplet on a CFX-96 Real-time PCR system (Bio-Rad, Hercules, CA, USA) (https://www.bio-rad.com) with SGExcel UltraSYBR Mixture (Sangon, Shanghai, China, https://www.sangon.com/). The PCR program was as follows: 95 °C for 3 min; 40 cycles of 95 °C for 15 s, 55 °C for 20 s, and 72 °C for 25 s. The primers used for qPCR are listed in Appendix A. The relative expression levels of genes were determined using the 2^−Δ*C*q^ method [55].

### 4.4. Library Construction and Sequencing

The samples were processed by Shanghai Majorbio Bio-pharm Technology Co. Ltd. (Shanghai, China) (http://www.majorbio.com/). A total of 1 μg RNA per sample was used as input material for RNA preparation. Finally, three independent samples for each treatment with RNA integrity number (RIN) value ≥ 8.0 were used for library construction and sequencing. Sequencing libraries were generated with a Truseq™ RNA sample prep Kit (Illumina, San Diego, CA, USA) (https://www.illumina.com/), and index codes were applied to identify each sample. Clusters of index-coded samples were generated on cBot Cluster Generation System (Illumina, San Diego, CA, USA) and library sequencing were performed on Illumina HiSeq 4000 platform for 150 bp paired-end reads generating. 5.5. Reads Filtration and Mapping.

### 4.5. Reads Filtration and Mapping

Raw data in fastq format were firstly processed with a fastx toolkit version 0.0.13 (The Cold Spring Harbor Laboratory, Long island, NY, USA) [56], and Q20, Q30, and GC contents were calculated. Clean data were obtained after trimming of adapter sequences, removing low quality reads, and filtering short reads using SeqPrep (https://github.com/jstjohn/SeqPrep) and Sickle (https://github.com/najoshi/sickle). All downstream analyses were based on high-quality clean data.

The clean data were mapped to the latest Glycine max genome assembly (Glycine_max_v2.1, Ensembl Plants) [57] by HISAT2 version 2.1.0 (Johns Hopkins University, Baltimore, MD, USA) [58] with default parameters. The quality of transcriptome was evaluated by RSeQC version 2.3.6 (Baylor College of Medicine, Houston, TX, USA) [59] according to mapped reads. 5.6. Quantification and Differential Expression of Genes.

### 4.6. Quantification and Differential Expression of Genes

Number of reads that were mapped to each gene was calculated with StringTie version 1.3.3 (Johns Hopkins University, Baltimore, MD, USA) [60], and normalized expression value TPM for genes was calculated by RSEM (University of Wisconsin-Madison, Madison, WI, USA) (http://deweylab.github.io/RSEM/). Differential gene expression analysis between each pair of samples was performed by using DESeq2 (European Molecular Biology Laboratory, Heidelberg, Germany) [61], and adjusted *p* value < 0.05 (false discovery rate (FDR) correction with Benjamini Hochberg method), and |log2FC| ≥ 1 were set as the criteria to filter the significant DEGs. DEGs clustering analysis was performed with Morpheus (The Broad Institute of MIT and Harvard Cambridge, MA, USA) (https://software.broadinstitute.org/morpheus).

### 4.7. Gene Ontology Analysis

Gene ontology terms were obtained from GO Consortium (Berkeley, CA, USA) [62] with BLAST2GO (Boston, MA, USA) [63], and the GO enrichment analysis were performed with Goatools [64] (FDR ≤ 0.05). Kyoto Encyclopedia of Genes and Genomes annotations were assigned according to the KEGG database (release 88.2; https://www.kegg.jp/), and DEGs in KEGG pathway were enriched with KOBAS version 3.0.0 (Peking University, Beijing, China) [65].

### 4.8. LC-MS Analysis of Phytohormones

To reveal the mechanism of aphid resistance in different soybean genotypes, endogenetic JA and SA content during aphid infestation were measured by LC-MS (SCIEX, Boston, MA, USA). Samples were harvested from three individual seedlings at 24, 48, and 96 HAI. About 0.15 g leaf tissue from the caged parts was ground with liquid nitrogen and used for phytohormone extraction with 1 mL of 80% (*v/v*) methanol. The extract was enriched with HyperSep C18 SPE chromatography (Thermo Scientific, Waltham, MA, USA) and concentrated under reduced pressure. Then, 10 μL of concentrate was used to detect hormone content by LC-MS (SCIEX, Boston, MA, USA) (https://sciex.com/) on Shimadzu LC-30AD (Shimadzu, Kyoto, Japan) (https://www.shimadzu.com/). The LC-MS conditions were as follows: a mobile phase of solvent A (acetonitrile) and solvent B (methanoic acid) (1/1000, *v/v*), a constant column temperature (35 °C). The detector components were Hybrid Quadrupole-TOF LC/MS/MS Mass Spectrometer (Shimadzu, Kyoto, Japan) and TripleTOF 5600+ (Shimadzu, Kyoto, Japan). The scanning ion for JA was *m*/*z* 211.1229–211.1429 and for SA m/z 139.0290–139.0490. The hormone content was quantified with an external standard (Shanghai Yuanyebio Bio-Technology Shanghai, China) (http://www.shyuanye.com/) as a reference. The results are presented as the mean of three biological replicates. 

### 4.9. Callose Deposition with Aphid Infestation

Callose deposition during aphid infestation was evaluated as described previously [66] with some modifications. Leaf samples from caged parts of individual seedlings were excised at desired time points and aphids were removed. Samples were fixed in formaldehyde-acetic acid-alcohol solution (FAA). After 24 h of fixation, FAA solution was removed, and the samples were washed with ethanol, sequentially incubated in 50% (*v/v*) ethanol and K_2_HPO_4_ buffer (pH 12) for 30 min, and then stained in the dark for 1 h with K_2_HPO_4_ buffer (pH 12) containing 0.01% aniline blue. After staining, the samples were mounted in 50% glycerol on a slide and observed using a UV2A filter (excitation wavelength = 330–380 nm) under an epifluorescence microscope (Axio Imager 2, Zeiss, Oberkochen, Germany) (https://www.zeiss.com/). Three biological replicate assays, each with at least 10 samples from independent seedlings, were prepared.

### 4.10. Reactive Oxygen Species Detection after Aphid Infestation

To investigate the possible function of ROS-related genes in aphid infestation, in situ detection of hydrogen peroxide with DAB staining was performed using an adaptation of a previous method [67]. Briefly, leaf samples from caged parts of individual seedlings were collected and aphids were removed. Samples were immersed in 1 mg/mL DAB containing Tween20 (0.05%, *v/v*) and 10 mM sodium phosphate buffer (pH 7.0) and infiltrated under a gentle vacuum for 5 min. After staining for 4 h in the dark, samples were fixed in bleaching solution (ethanol: acetic acid: glycerol = 3:1:1), and placed in water bath at 95 °C for 15 min. Samples were kept in bleaching solution till chlorophyll was completely depleted and then visualized and photographed under white light (Olympus MVX10, Olympus, Tokyo) (https://www.olympus-lifescience.com/en/). ImageJ [68] was used to observe and quantify the DAB staining area. Three biological replicate assays, each with at least 10 samples from independent seedlings, were performed.

### 4.11. Statistical Analysis

Phytohormones content, callose deposition, and ROS detection were analyzed by one-way analysis of variance. Separation of the means was performed using Fisher’s protected least significance difference test at *p* = 0.05. Statistical analyses were performed with SPSS v16.0 (IBM Corp, Armonk, NY, USA) (https://www.ibm.com/products/spss-statistics).

## 5. Conclusions

Transcriptome profiles of susceptible, antibiotic, and antixenotic genotypes in soybean with aphid infestation were revealed. The response of antibiotic genotype was faster than susceptible genotype during early infestation, while the plant-aphid interaction was less in antixenotic genotype. Susceptible and antixenotic genotypes was similar in the response to aphid infestation in TF expression patterns, JA behavior, and callose deposition, while, in antibiotic genotype, SA was involved by suppressing the JA pathway, and callose deposition occurred more rapidly and efficiently. Besides, ROS was not involved in the response to aphid attack in resistant soybean.

## 6. Data Availability Statement

All datasets have been deposited to the Gene Expression Omnibus (GEO) database under accession number GSE141720.

## Figures and Tables

**Figure 1 ijms-21-05191-f001:**
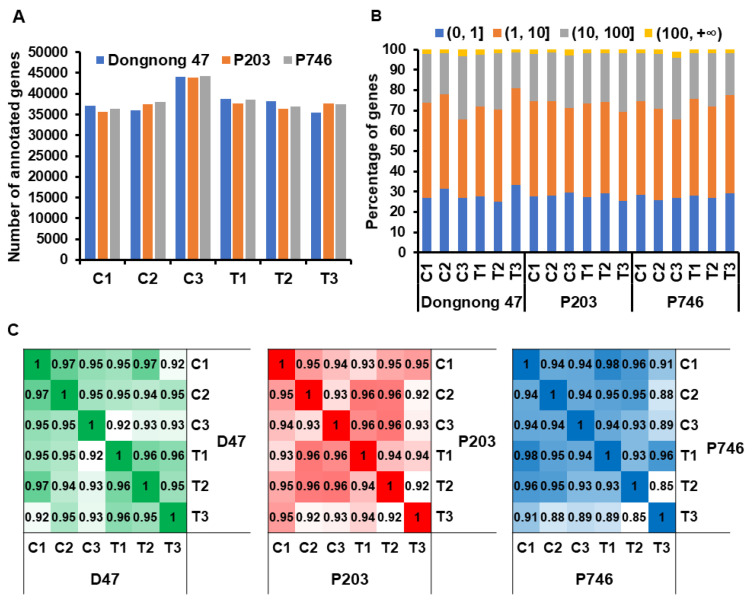
Overview of the gene expression in soybean cultivars Dongnong47, P203, and P746 with aphid attack in 24 h, 48 h, and 96 h after infestation. (**A**) Total number of annotation genes in the genotypes. (**B**) Proportion of genes with indicated expression strength at each level (transcripts per million reads, TPM). (**C**) Pearson correlation coefficients (PCCs) of gene expression (TPM) between different treatment. The PCCs of expressed genes from soybean genotypes Dongnong47, P203, and P746 were calculated separately.

**Figure 2 ijms-21-05191-f002:**
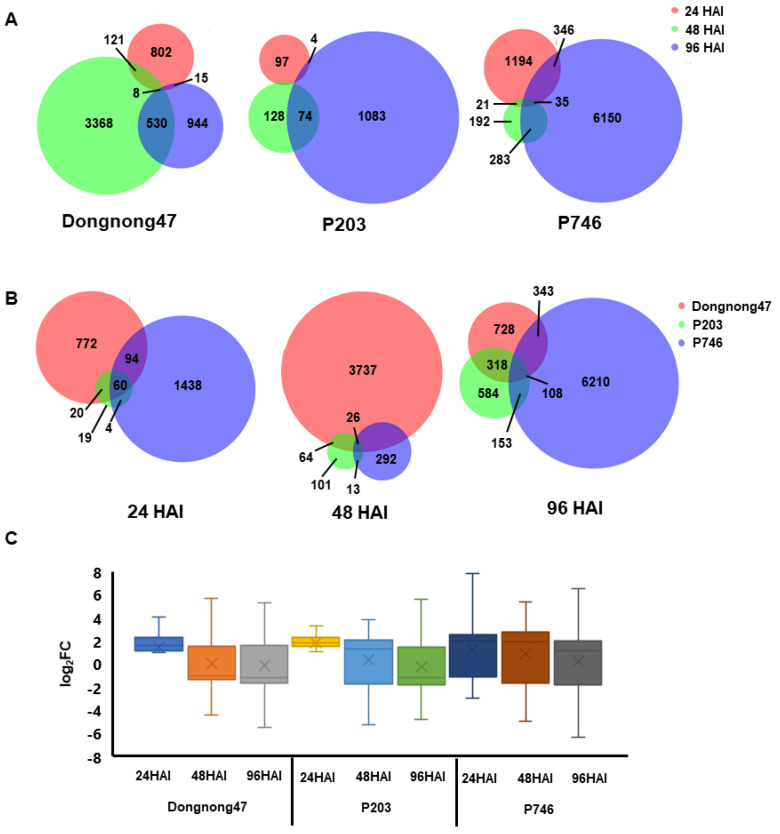
The differential expression genes in all genotypes. (**A**) Venn diagram of differentially expressed genes (DEGs) during aphid infestation in each genotype. (**B**) Venn diagram of DEGs from all genotypes at 24, 48, and 96 h after incubation (HAI), respectively. (**C**) The expression profiles of DEGs in all genotypes.

**Figure 3 ijms-21-05191-f003:**
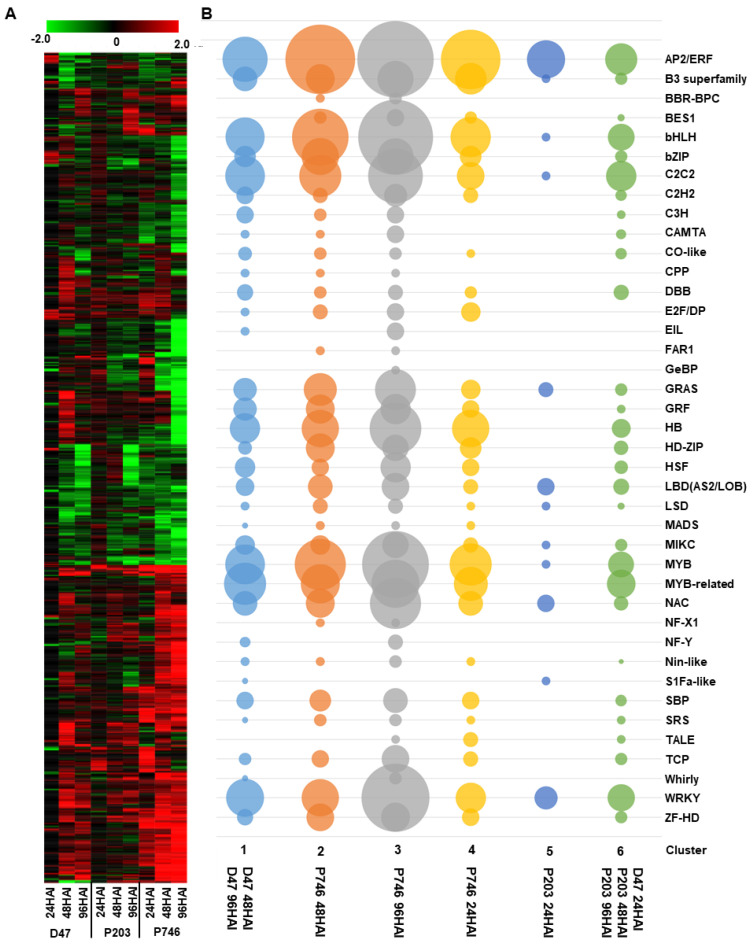
Different expressed transcription factors (TF). (**A**) Heatmap showing the expression level of DEGs in all genotypes. (**B**) TFs were clustered into different families and are listed on the right, and the size of the circle indicated the number of TFs.

**Figure 4 ijms-21-05191-f004:**
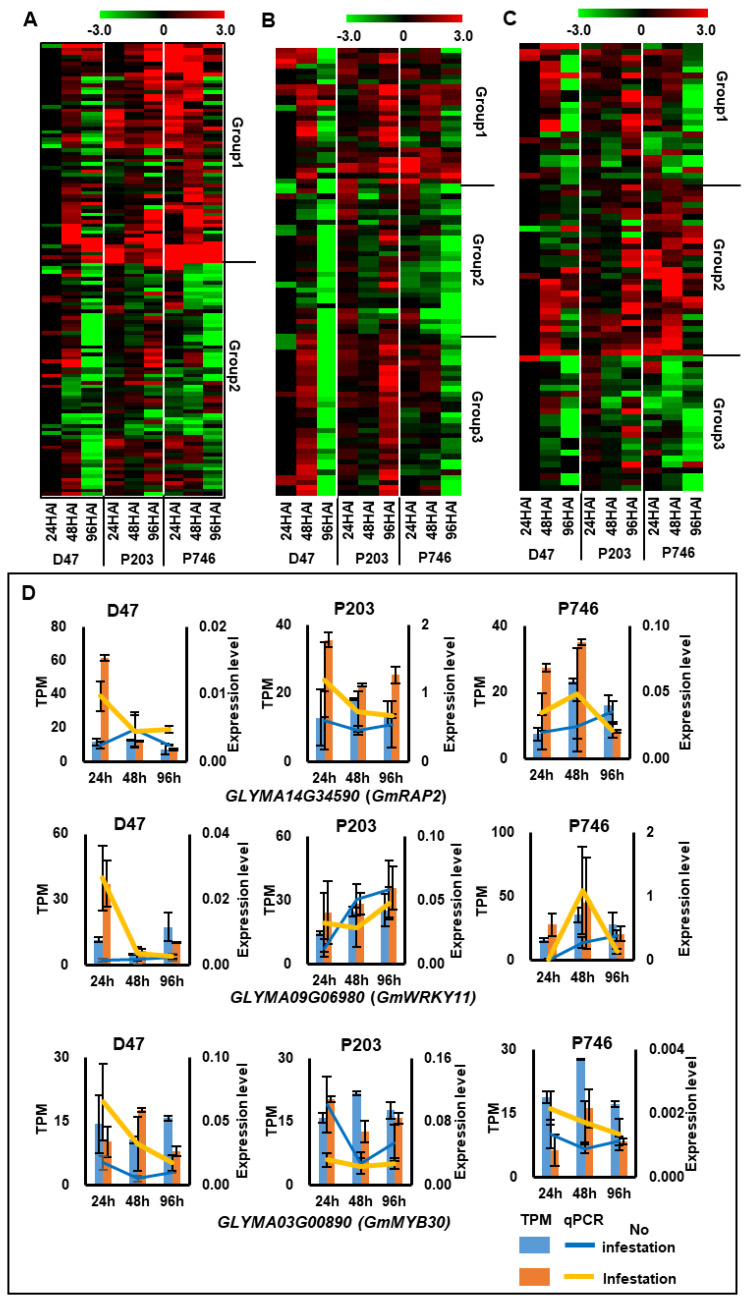
Expression patterns of APETALA 2/ethylene response factors (AP2/ERF), the transcription factors contained conserved WRKYGQK domain (WRKY), and v-myb avian myeloblastosis viral oncogene homolog (MYB) TFs. (**A**)–(**C**) Heat maps visualizes the expression patterns of AP2/ERF (A), WRKY (**B**), and MYB (**C**) TFs, qPCR validation of TF expression patterns (**D**): *GLYMA14G34590 (GmRAP2)*, *GLYMA09G06980 (GmWRKY11)*, *GLYMA03G00890 (GmMYB30)*. Columns show the TPM according to RNA-seq; lines indicate expression levels in qPCR by 2^−Δ*C*q^.

**Figure 5 ijms-21-05191-f005:**
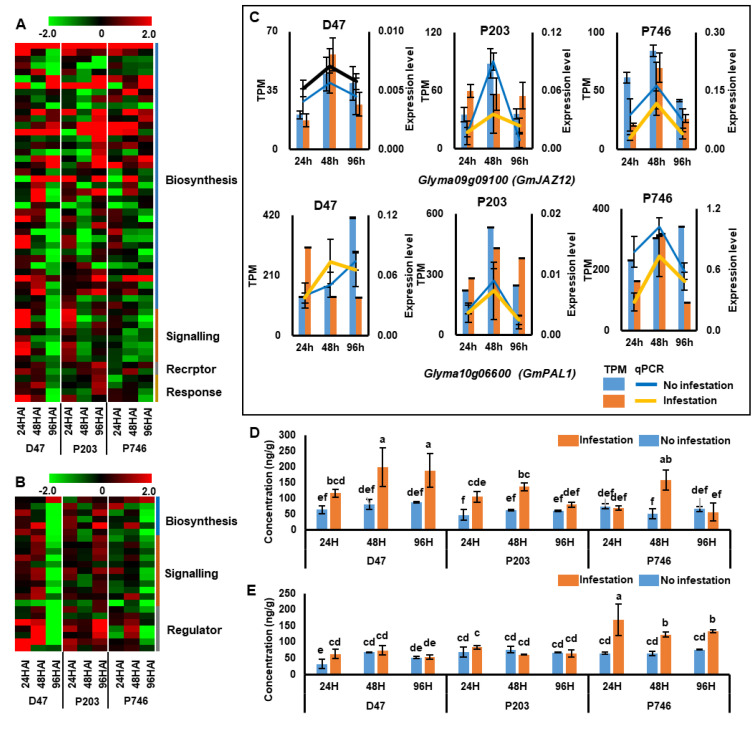
Expression patterns of genes related to jasmonic acid (JA) and salicylic acid (SA). (**A**)–(**B**) Heat maps visualize the expression patterns of JA- and SA-related genes. (**C**)–(**E**) qPCR validation of gene expression patterns. Columns show the transcripts per million reads according to RNA-seq; lines indicate expression levels in qPCR according to 2^−ΔCq^ method. Concentration of JA (**D**) and SA (**E**) in all situations were evaluated. Different letters indicate significant difference (*n* = 3, *p* < 0.05).

**Figure 6 ijms-21-05191-f006:**
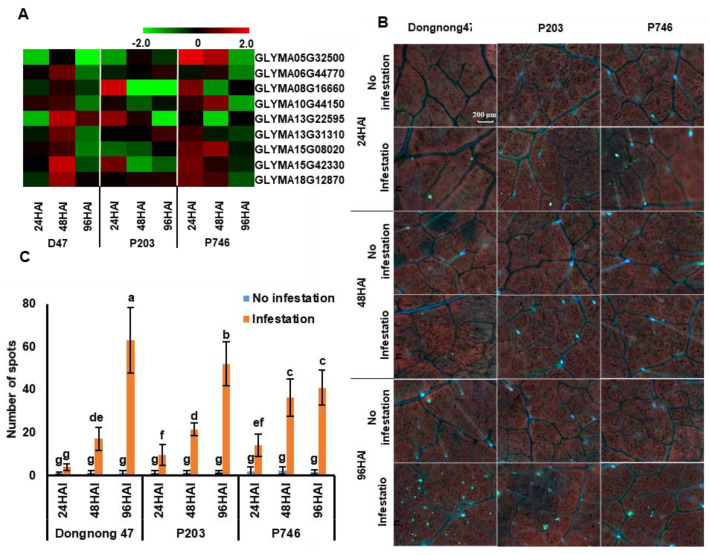
Expression patterns of callose synthase genes during aphid infestation. (**A**) Changes in the expression of genes related to callose synthase (*CalS*) according to RNA-seq data. (**B**) Callose deposition was detected with aniline blue staining. (**C**) Changes in callose deposition evaluated with number of spots with florescence. Callose deposition was calculated in 0.5 mm^2^ microscopic fields for each leaf. Different letters indicate significant difference (*n* = 10, *p* < 0.05). Three biological replicate assays, each with at least 10 samples from independent seedlings, were performed.

**Figure 7 ijms-21-05191-f007:**
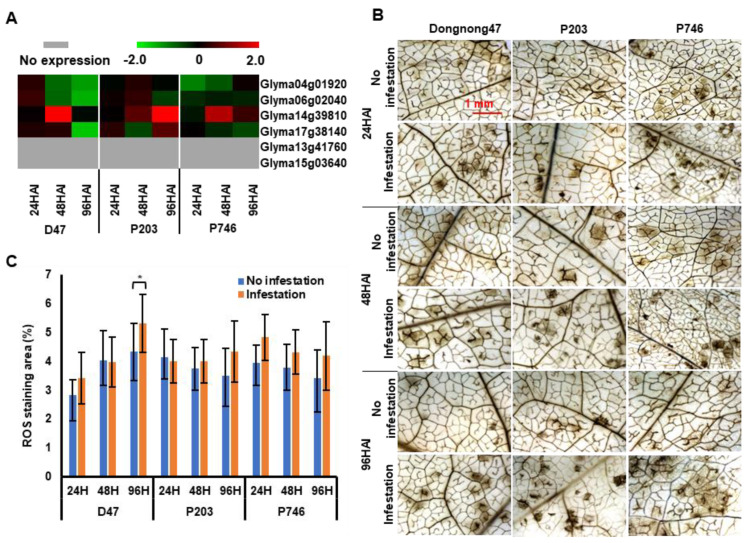
Expression patterns of catalase-related genes. (**A**) Changes in the expression of genes related to catalase according to RNA-seq data. (**B**) The activity of hydrogen peroxide was detected using in situ detection with 3,3′-diaminobenzidine staining (DAB). (**C**) The activity of hydrogen peroxide was evaluated as the percentage of staining calculated in 0.5 mm^2^ microscopic fields for each leaf. Asterisk * indicate significantly higher values of ROS staining area at 96 h with aphid infestation in D47. Several results were also observed in resistant or susceptible genotypes of other plants under inse (*n* = 10, *p* < 0.05). Three biological replicate assays, each with at least 10 samples from independent seedlings, were performed.

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
