# Peer review of "A Genome-Wide View of Transcriptional Responses during Aphis glycines Infestation in Soybean"

_ijms, 2020, doi:10.3390/ijms21155191_

Round 1

Reviewer 1 Report

This MS on 'genome-wide view of transcriptional responses during Aphis glycines infestation in soybean' presents some valuable results in the differential expression of genes and different pathways. I have my recommendation to publish this MS with minor revisions. 

This MS presents the trancriptional responses after 24, 48, 96 hrs of infestation, but didn't study the immediate plant response after aphid infestation which could provide the better comparison of plant responses for the initiation resistance response among the genotypes having susceptible and different resistance mechanism.  

The MS presents the various differential expressions among the genotypes of different R mechanism with susceptible. Authors also tried to discuss about what those differential expression but I feel that the discussion could be more improved to give more conclusive explanation.

Overall, It's a good MS.   

Author Response

Response to Reviewer 1 Comments

Point 1: The discussion could be more improved to give more conclusive explanation.

Response 1: Thanks for your suggestion. We have added the description as follows at line 235:

Several results were also observed in resistant or susceptible genotypes of other plants under insect attack or bacterial infection, which indicated the general response of plant to biotic stress.

Also, the section of Conclusion was added to conclude and emphasize results as follows at line 334:

Transcriptome profiles of susceptible, antibiotic and antixenotic genotypes in soybean with aphid infestation were revealed. The response of antibiotic genotype was faster than susceptible genotype during early infestation, while the plant-aphid interaction was less in antixenotic genotype. Susceptible and antixenotic genotypes was similar in the response to aphid infestation in TF expression patterns, JA behavior and callose deposition, while in antibiotic genotype SA was involved by suppressing the JA pathway and callose deposition occurred more rapidly and efficiently. Besides, ROS was not involved in the response to aphid attack in resistant soybean.

Reviewer 2 Report

I have read this manuscript several times and it is overall very easy to read and understand. The authors have done a huge amount of work, which has been presented in throughout explained tables and figures. Congratulations for that! The figures are very nice and easy to understand. As a minor comment, because the authors have many data, I had a harsh time in searching the SI files. In this context, I think it would help the reader if some of the information currently deposited in SI files could be inside the main manuscript.

I also have some further comments:

Introduction:

  • Please include more details about the introduction of aphid and why it become invasive.
  • Please include information about the influence of the aphid on reducing photosynthesis.
  • Please give more details about antibiosis and antixenosis. This is crucial to understand the aims of the study and the findings found.

Material and Methods:

  • Details about the cultivars used are missing. How many different plants were used and from how many seeds?
  • How many samples were extracted?

Results

  • Clean reads were mapped to chromosomes. How was this done?
  • Fig. S2 is central do understand DEGs. It should be presented inside the manuscript.
  • Same for the functional analysis. A resume of Figures S3 and S4 should be presented. Like it is presented, I don't see the importance of TFs (in comparison to other genes).
  • I am not sure about the importance of the results concerning phytohormones, callose deposition or ROS genes. I am not saying that the authors should eliminate those results; what I am saying is that the reader reads these results without any previous introduction of why this was done. Like it is written, the reader has a harsh time in understanding this because most previous information comes from several tables and figures that are SI files.

Discussion

  • Are these results important to other crops infested by aphids or is this only relevant to soybean?

Author Response

Response to Reviewer 2 Comments

Point 1: I think it would help the reader if some of the information currently deposited in SI files could be inside the main manuscript.

Response 1: Thanks for your suggestion. We have moved Figure S2 to the main manuscript as Figure 2 at line 171 and modified Figure S3, S4.

Introduction:

Point 2: Please include more details about the introduction of aphid and why it become invasive.

Response 2: Thanks for your suggestion. We have added more details as follows at line 30:

Soybean aphid, Aphis glycine (Matsumura), is an asexually-reproducing soybean pest native to eastern Asia, and after its accidental introduction in 2000, soybean aphid became invasive throughout Midwestern US and southern Canada because of the rapid population expansion, high mobility and few limits to migration.

Point 3: Please include information about the influence of the aphid on reducing photosynthesis.

Response 3: Thanks for your suggestion. We have added more details as follows at line 33:

Soybean aphid infestation reduces soybean yield by removing photosynthates and inhibiting photosynthesis up to 50% on infested leaves even without apparent symptoms of aphid injury.

Point 4: Please give more details about antibiosis and antixenosis.

Response 4: Thanks for your suggestion. We have added more details as follows at line 44:

Antibiotic mechanisms influence aphid physiology, such as development time, survival, and fecundity by toxic secondary metabolites, protease inhibitors and antibiotic effects; antixenotic factors affect aphid behavior, such as plant choice and feeding behavior by volatiles and physical barrier of plant.

Material and Methods:

Point 5: Details about the cultivars used are missing. How many different plants were used and from how many seeds?

Response 5: Thanks for your suggestion. We have added the information as follows at line 348:

At least 18 seedlings of each genotype were retained for treatment with one seedling per pot.

Point 6: How many samples were extracted?

Response 6: Thanks for your suggestion. We have added the information as follows at line 364:

Three independent samples for each treatment were harvested for RNA extraction.

Results

Point 7: Clean reads were mapped to chromosomes. How was this done?

Response 7: Thanks for your suggestion. We have added the information as follows at line 82:

The distribution of clean reads on chromosomes was analyzed to estimate the distribution of genes responding to aphid infestation (Table S2) according to the latest soybean genome Wm82.a2.v1 from the Ensembl Plants database (https://plants.ensembl.org/index.html).

Point 8: Fig. S2 is central do understand DEGs. It should be presented inside the manuscript.

Response 8: Thanks for your suggestion. We have moved the figure into the manuscript as Figure 2 at line 171.

Point 9: A resume of Figures S3 and S4 should be presented. Like it is presented, I don't see the importance of TFs (in comparison to other genes).

Response 9: Thanks for your suggestion. We have modified the figures and labeled them as Figure S2 and S3 respectively. And more description related to TFs were added as follows at line 115:

The enriched GO terms suggested that transcription regulators were involved and play key roles during aphid infestation.

Point 10: I am not sure about the importance of the results concerning phytohormones, callose deposition or ROS genes. I am not saying that the authors should eliminate those results; what I am saying is that the reader reads these results without any previous introduction of why this was done. Like it is written, the reader has a harsh time in understanding this because most previous information comes from several tables and figures that are SI files.

Response 10: Thanks for your suggestion. We have added related information in the section of GO enrichment analysis as follows at line 116:

Besides, genes associated to phytohormones regulation (such as JA and SA biosynthetic and metabolic process), defense response related to callose (such as callose localization and deposition in cell wall) and superoxide metabolism (such as superoxide radicals removal) were also enriched (Table S5)

Discussion

Point 11: Are these results important to other crops infested by aphids or is this only relevant to soybean?

Response 11: Thanks for your suggestion. We have added the description as follows at line 235:

Several results were also observed in resistant or susceptible genotypes of other plants under insect attack or bacterial infection, which indicated the general response of plant to biotic stress.

Also, the section of Conclusion was added to conclude and emphasize results as follows at line 334:

Transcriptome profiles of susceptible, antibiotic and antixenotic genotypes in soybean with aphid infestation were revealed. The response of antibiotic genotype was faster than susceptible genotype during early infestation, while the plant-aphid interaction was less in antixenotic genotype. Susceptible and antixenotic genotypes was similar in the response to aphid infestation in TF expression patterns, JA behavior and callose deposition, while in antibiotic genotype SA was involved by suppressing the JA pathway and callose deposition occurred more rapidly and efficiently. Besides, ROS was not involved in the response to aphid attack in resistant soybean.